# Loss of a co-twin at birth and subsequent risk of psychiatric disorders

Huan Song[1,2,3]*, Fang Fang[4], Henrik Larsson[3,5], Nancy L Pedersen[3,6], Patrik KE Magnusson[3], Catarina Almqvist[3,7], Unnur A Valdimarsdóttir[2,3,8]

[1]West China Biomedical Big Data Center, West China Hospital, Sichuan University, Sichuan, China; [2]Center of Public Health Sciences, Faculty of Medicine, University of Iceland, Reykjavík, Iceland; [3]Department of Medical Epidemiology and Biostatistics, Karolinska Institutet, Stockholm, Sweden; [4]Institute of Environmental Medicine, Karolinska Institutet, Stockholm, Sweden; [5]School of Medical Sciences, Örebro University, Örebro, Sweden; [6]Department of Psychology, University of Southern California, Los Angeles, United States; [7]Astrid Lindgren Children's Hospital, Karolinska University Hospital, Stockholm, Sweden; [8]Department of Epidemiology, Harvard T H Chan School of Public Health, Boston, United States

**Abstract** Twins suffering a co-twin loss at birth have reported feelings of loneliness and grief while it remains unexplored if they suffer increased risk of psychiatric disorders. We contrasted rate of first-onset psychiatric disorders among all Swedish-born twins whose co-twin died within 60 days after birth between 1973 and 2011 (n = 787) to that of 3935 matched unexposed twins, 3935 matched singletons (both matched to the exposed twins by birth year, sex, and birth characteristics), and 880 full siblings of the exposed twins. During a median of 19-year follow-up, exposed twins were at increased risk of first-onset psychiatric disorders (hazard ratio = 1.56, 95%CI 1.30–1.87) compared with unexposed twins. We observed the strongest association for emotional disorders and for psychiatric disorders diagnosed before the age of 25. Comparisons with matched singletons and the twin's full siblings rendered similar results, corroborating an association of loss of a co-twin at birth with subsequent risk of psychiatric disorders.

*For correspondence:
songhuan@wchscu.cn

Competing interests: The authors declare that no competing interests exist.

## Introduction

Because of the sharp rise in medically assisted reproduction and delayed childbearing during the last decades, the twinning rate has increased dramatically in all developed countries (*Pison et al., 2015*). This rise in twinning rate represents an important public health issue since twin pregnancies are associated with greater health risks for both infants and mothers (*Cheong-See et al., 2016*). Compared to singletons, twin babies are more often subjected to multiple adverse neonatal outcomes, including preterm birth, small-for-gestational-age, and neonatal death (*Cheong-See et al., 2016*). In addition, twin pregnancies are further complicated with discordant growth (*Miller et al., 2012*; *Grantz et al., 2016*; *D'Antonio et al., 2018*). Approximately 16% of twin pregnancies have birth weight discordance of at least 20%, which might also contribute to elevated risk of perinatal and neonatal mortality (*Blickstein and Kalish, 2003*). Consequently, a considerable proportion of twins experience a very early loss of their co-twin.

Previous studies indicate that loss of a co-twin by death in childhood or adulthood is associated with considerable mental morbidities among the surviving twins (*Segal and Bouchard, 1993*; *Woodward, 1988*). Our recent findings suggest an increased risk of psychiatric disorders after a loss of co-twin, compared to loss of a full (non-twin) sibling, beyond age 2 (*Song et al., 2020*). The greater risk increase after loss of a co-twin might be due to the stronger emotional bond between twins (*Rosendahl SP and Björklund, 2013*; *Segal and Ream, 1998*) and the greater genetic relatedness,

in the case of monozygotic twins (*Segal and Ream, 1998*; *Parkes, 1993*; *Neyer, 2002*). In contrast, with cognitive immaturity and limited (or no) afterbirth interactions, a co-twin loss at birth or during the neonatal period leaves little room for a twin bond to be established and therefore it seems implausible that the mental health of the surviving twin would be affected by such a loss. However, several scientific and media accounts describe unexpected lingering sorrow among twins who lost their co-twin at or shortly after birth, even among the twins that didn't know they were born as twins (*Morgan, 2014*; *lone twin network, 2020*; *Woodward, 2010*). It is further possible that the parenting of bereaved parents after the perinatal loss of one twin baby may leave the surviving twin vulnerable for mental morbidities (*Lamb, 2002*). In the total absence of data on the rate of psychiatric disorders among twins who lost a co-twin at birth, we conducted a nationwide population- and sibling-matched cohort study to estimate the extent to which loss of a co-twin at birth is associated with the incidence of psychiatric disorders among surviving twins, after carefully controlling for important confounders such as birth characteristics and familial factors.

## Results

In a population-based matched cohort, we included in the study all Swedish-born twins that lost a co-twin at birth, defined as a death of the co-twin within 60 days after birth, between 1973 and 2011- (exposed twins, n = 787), together with two reference groups. The first reference group included 3935 sex-, birth year-, and gestational age (GA)-matched unexposed twins randomly selected from the Swedish twin population that did not experience such a loss, to control for twin pregnancy and twin birth. Because the exposed twins grew up on their own, as singletons, which differs from the social conditions of a twin life, we also included 3935 singletons randomly selected from the singleton population that were individually matched to the exposed twins on sex, birth year, and birth factors (GA, birth weight for GA, and birth order) as the second reference group. In addition to the population-based matched cohort, to address the concern of familial factors such as genetic background and environmental factors during childhood shared within a family (e.g., parenting of the bereaved parents), we compared 569 exposed twins to their full siblings (n = 880) in twin-sibling family cohort. The study entry was the 60th day after birth of the exposed twin (i.e., the index date). We then followed all individuals from the index date until the first diagnosis of any psychiatric disorder, emigration, death, or the end of 2013, whichever occurred first (see details in the later section 'Materials and methods,' and *Figure 1* and *Supplementary file 1*-Table 1).

In total, the population-based matched cohort accumulated 169,507 person-years at risk, with a median of 19 years of follow-up (*Table 1*). The exposed twins had an almost equal sex distribution (54% were male). While there was little difference in birth weight for GA, maternal educational level, and family history of psychiatric disorders, exposed twins tended to have lower Apgar score, older mothers at childbirth, and higher maternal cohabitation rate than matched unexposed twins and singletons. As expected, in the twin-sibling family cohort, maternal characteristics were identical or similar between exposed twins and their full siblings, although birth characteristics (e.g., GA, birth weight for GA, and Apgar score) were different between these two groups (*Table 1*).

During the follow-up, we identified 1501 individuals with incident psychiatric disorders in the population-based matched cohort, including 178 cases among the exposed twins, 600 among the matched unexposed twins, and 723 among the matched singletons, corresponding to a crude incidence rate (IR) of 12.08, 7.76, and 9.33 per 1000 person-years, respectively (*Table 2*). Compared to matched unexposed twins, the rate of any psychiatric disorders was increased among the exposed twins. The hazard ratio (HR) was 1.59 (95% confidence intervals [CI] 1.33–1.90) after controlling for birth year, sex, and birth factors (i.e., GA, birth weight for GA, maternal age at birth), and decreased to 1.56 (95% CI 1.30–1.87) when other covariables, including Apgar score, family history of psychiatric disorders, and maternal educational level and cohabitation status were added into the model. The fully adjusted HR was 1.41 (95% CI 1.19–1.69) when the reference group was matched singletons. In the analyses of twin-sibling family cohort, we obtained an HR of 1.43 (95% CI 0.82–2.51) after full adjustment of all abovementioned variables (*Table 2*).

Although not statistically significant, we observed higher HRs for emotional disorders such as depression and anxiety, compared with neurodevelopmental disorders and other psychiatric disorders (*Table 2*). Subgroup analyses of the population-based matched cohort indicated that the association between loss of a co-twin at birth and subsequent risk of any psychiatric disorder did not

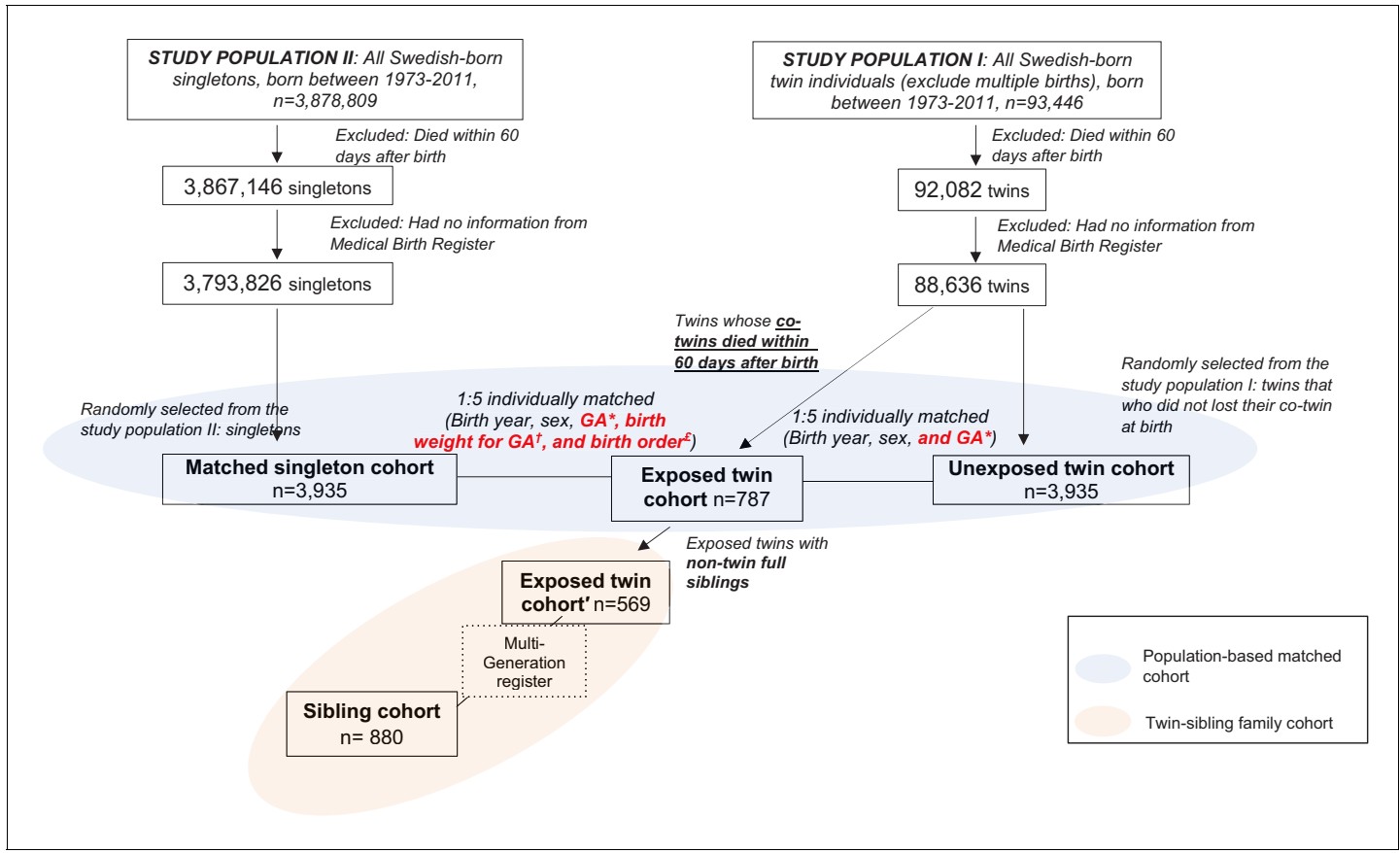

**Figure 1.** Study design. *GA, Gestational age, categorized as <28 week, 28–31 weeks, 32–36 weeks, >36 weeks. †Birth weight for gestational age was generated by calculating birth weight z-score for each gestational age and sex-specific population, and was categorized as <10th, 10–30th, 31–50th, 51–70th, 71–90th, >90th, or unknown. We did this separately for twins and singletons. £Birth order was categorized as 1, 2, 3, ≥4.

differ by sex, but seemed stronger among individuals without family history of psychiatric disorders. In addition, the relative risk increased further after loss of a same-sex co-twin, than loss of an opposite-sex co-twin, at birth (*Table 3*). By plotting HRs over attained age, we found the risk elevation to be only evident for psychiatric disorders diagnosed in childhood and early adulthood, that is before age of 25 (*Figure 2*). With lower precision, analyses of the twin-sibling family cohort revealed similar risk patterns (*Table 3*). We obtained slightly higher HRs when comparing the exposed twins to their older siblings (1.74, 95% CI 0.54–5.59), than to their younger siblings (1.27, 95% CI 0.48–3.34) (*Supplementary file 1*-Table 2).

In sensitivity analyses, we found no significant effect modification by the presence of psychiatric disorders among parents (*Supplementary file 1*-Table 2), by the presence of congenital abnormalities, or by the diagnoses of severe somatic diseases during follow-up (*Supplementary file 1*-Table 3). Moreover, changing the definition of loss of a co-twin at birth to loss of a co-twin within 28 days after birth yielded largely similar estimates, although with lower precision (*Supplementary file 1*-Table 4).

## Discussion

To our knowledge, this is the first nationwide population-based and sibling-matched cohort study exploring the association between loss of a co-twin at birth or shortly after birth and the subsequent risk of psychiatric disorders. Compared to matched unexposed twins or singletons, as well as their non-twin full siblings, twins exposed to a co-twin loss at birth were at considerably elevated risk of psychiatric disorders, especially emotional disorders, such as depression and anxiety, before age of 25. Notably, this association was independent of multiple important confounders, including birth

**Table 1.** Characteristics of the study cohorts.

| | Population-based matched cohort: twins exposed to loss of co-twin at birth vs. unexposed twins or singletons | | | Twin-sibling family cohort: twins exposed to loss of a co-twin at birth vs. their full siblings | |
|---|---|---|---|---|---|
| | Exposed twins | Matched unexposed twins | Matched singletons | Exposed twins | Exposed full siblings |
| Number of individuals | 787 | 3935 | 3935 | 569 | 880 |
| Age at end of follow-up, median (interquartile range), year | 18.0 (9.5–28.0) | 19.8 (10.7–29.2) | 19.5 (10.8–28.9) | 18.3 (9.7–26.7) | 20.5 (11.8–28.0) |
| Follow-up time, median (IQR), year | 17.8 (9.3–27.8) | 19.3 (10.1–28.7) | 19.0 (10.3–28.5) | 18.1 (9.5–26.5) | 18.6 (9.9–25.1) |
| % of male | 53.9 | 53.8 | 53.9 | 55.7 | 52.7 |
| Gestational age, n (%) | | | | | |
| <28 weeks | 181 (23.0) | 814 (20.7) | 833 (21.2) | 142 (25.0) | 2 (0.23) |
| 28–31 weeks | 184 (23.4) | 1001 (25.4) | 1052 (26.7) | 139 (24.4) | 6 (0.68) |
| 32–36 weeks | 216 (27.5) | 1090 (27.7) | 1030 (26.2) | 151 (26.5) | 62 (7.05) |
| ≥37 weeks | 162 (20.6) | 810 (20.6) | 800 (20.3) | 132 (23.2) | 809 (91.9) |
| Unknown | 44 (5.59) | 220 (5.59) | 220 (5.59) | 5 (0.88) | 1 (0.11) |
| Birth weight for gestational age*, n (%) | | | | | |
| <10th | 82 (10.4) | 287 (7.29) | 417 (10.6) | 59 (10.4) | 76 (8.64) |
| 10–30th | 156 (19.8) | 738 (18.8) | 774 (19.7) | 113 (19.9) | 151 (17.2) |
| 31–50th | 207 (26.3) | 908 (23.1) | 1031 (26.2) | 151 (26.5) | 173 (19.7) |
| 51–70th | 136 (17.3) | 832 (21.1) | 681 (17.3) | 101 (17.8) | 175 (19.9) |
| 71–90th | 108 (13.7) | 589 (15.0) | 536 (13.6) | 78 (13.7) | 194 (22.1) |
| >90th | 54 (6.86) | 361 (9.17) | 276 (7.01) | 39 (6.9) | 108 (12.3) |
| Unknown | 44 (5.59) | 220 (5.59) | 220 (5.59) | 28 (4.9) | 3 (0.34) |
| Apgar score ≤ 7 at 5/10 min, n (%) | | | | | |
| No | 567 (72.1) | 3121 (79.3) | 3222 (81.9) | 411 (72.2) | 838 (95.2) |
| Yes | 118 (15.0) | 332 (8.44) | 392 (9.96) | 84 (14.8) | 9 (1.02) |
| Unknown | 102 (13.0) | 482 (12.3) | 321 (8.16) | 74 (13.0) | 33 (3.75) |
| Maternal age at birth, n (%) | | | | | |
| ≤28 | 345 (43.8) | 1788 (45.4) | 2179 (55.4) | 267 (46.9) | 397 (45.1) |
| 29–32 | 209 (26.6) | 1098 (27.9) | 912 (23.2) | 158 (27.8) | 245 (27.8) |
| ≥33 | 233 (29.6) | 1049 (26.7) | 844 (21.5) | 144 (25.3) | 238 (27.1) |
| Maternal educational level, n (%) | | | | | |
| <9 years | 36 (4.57) | 190 (4.83) | 178 (4.52) | 19 (3.34) | 31 (3.52) |
| 9–12 years | 498 (63.3) | 2418 (61.5) | 2618 (66.5) | 359 (63.1) | 569 (64.7) |
| >12 years | 241 (30.6) | 1272 (32.3) | 1089 (27.7) | 185 (32.5) | 273 (31.0) |
| Unknown | 12 (1.52) | 55 (1.40) | 50 (1.27) | 6 (1.05) | 7 (0.80) |
| Maternal cohabitation status, n (%) | | | | | |
| Yes | 671 (85.3) | 3165 (80.4) | 3067 (77.9) | 494 (86.8) | 783 (89.0) |
| No | 112 (14.2) | 748 (19.0) | 847 (21.5) | 72 (12.7) | 90 (10.2) |
| Unknown | 4 (0.51) | 22 (0.56) | 21 (0.53) | 3 (0.5) | 7 (0.80) |
| Family history of psychiatric disorders including suicide, n (%) | | | | | |
| Yes | 78 (9.91) | 334 (8.49) | 395 (10.0) | 49 (8.61) | 93 (10.6) |
| No | 709 (90.1) | 3601 (91.5) | 3540 (90.0) | 520 (91.4) | 787 (89.4) |

* Birth weight was standardized by singletons/twins, sex, and gestational age.

**Table 2.** Hazard ratios (HRs) with 95% confidence intervals (CIs) for any psychiatric disorder among twins after loss of a co-twin at birth, derived from different Cox models and by subtypes of psychiatric disorders.

| | Population-based matched cohort | | | | Twin-sibling family cohort | |
| --- | --- | --- | --- | --- | --- | --- |
| | Number of cases (crude incidence rate, per 1000 person years), exposed twins/unexposed twins | HR (95% CI)[*] | Number of cases (crude incidence rate, per 1000 person years), exposed twins/matched singletons | HR (95% CI)[*] | Number of cases (crude incidence rate, per 1000 person years), exposed twins/full siblings | HR (95% CI)[*] |
| Model information: Model 1 Controlled for attained age, (as underlying time scale), sex, and birth characteristics (i.e., GA, birth weight for GA, maternal age at birth) | 178 (12.08)/600 (7.76) | 1.59 (1.33–1.90) | 178 (12.08)/723 (9.33) | 1.42 (1.19–1.68) | 130 (12.32)/130 (8.17) | 1.44 (0.83–2.51) |
| Model 2 above + neonatal factors (Apgar score) | | 1.57 (1.31–1.87) | | 1.37 (1.15–1.63) | | 1.43 (0.82–2.49) |
| Model 3 above + family history of psychiatric disorders, maternal educational level, maternal cohabitation status | | 1.56 (1.30–1.87) | | 1.41 (1.19–1.69) | | - |
| Full adjusted HRs[†] for subtypes of psychiatric disorders | | | | | | |
| Neurodevelopmental disorders (ADHD, ASD, and intellectual disabilities) | 71 (4.53)/224 (2.80) | 1.56 (1.16–2.08) | 71 (4.53)/270 (3.34) | 1.44 (1.09–1.92) | 52 (4.58)/35 (2.10) | 0.24 (0.05–1.30) |
| Emotional disorders (depression, anxiety, stress-related disorders) | 105 (6.75)/293 (3.67) | 1.90 (1.49–2.42) | 105 (6.75)/386 (4.82) | 1.57 (1.25–1.98) | 79 (7.04)/85 (5.21) | 1.75 (0.89–3.44) |
| Other psychiatric disorders | 84 (5.43)/310 (3.92) | 1.32 (1.02–1.70) | 84 (5.43)/361 (4.54) | 1.28 (1.00–1.64) | 60 (5.39)/61 (3.72) | 1.37 (0.61–3.08) |

ADHD, attention deficit hyperactivity disorder; ASD, autism spectrum disorder; GA, gestational age.

[*]Cox regression models were stratified by matching identifiers or family identifier, and adjusted for covariates mentioned in the 'model information' column. Attained age was applied as the underlying time scale.

[†]HRs were derived from fully adjusted Cox regression models, that is, model 3.

characteristics and childhood social conditions (by comparing bereaved twins to singletons), and other familial factors (by comparing bereaved twins to their full siblings), indicating that increased clinical alertness of the mental health of surviving twins after very early co-twin loss is warranted. In addition, although the excess risk was not modified by parent's psychiatric disorder nor the surviving twin's congenital or other severe diseases diagnosed during follow-up, it seemed more pronounced among twins exposed to early loss of a same-sex co-twin and among twins without family history of psychiatric disorders.

While accumulating evidence supports that both childhood and adult twin loss are associated with increased risk of psychiatric morbidity among the surviving twins (*Rosendahl SP and Björklund, 2013*; *Segal and Ream, 1998*), no previous study has addressed whether such emotional reactions can be observed after a very early co-twin loss where limited twin relationship, perception, or memory from the loss could be expected. The absence of evidence is mainly due to the complexity of the research question and lack of high-quality data to address potential confounding by multiple factors, such as twin pregnancy and birth (i.e., suboptimal birth characteristics) but a singleton-like life, familial factors, and genetic susceptibility to diseases. Therefore, with the unique Swedish nationwide data sources, which provide a substantial sample size of exposed twins with detailed data on birth characteristics and familial information, we conducted the present study. By contrasting the rate of psychiatric disorders among surviving twins who were exposed to a co-twin loss at birth with that of several comparison groups, including matched unexposed twins and singletons, as well as the full

**Table 3.** Hazard ratios (HRs) with 95% confidence intervals (CIs) for any psychiatric disorder among the surviving twins after co-twin loss at birth, by characteristics of the twin pairs.

| | Population-based matched cohort | | | | Twin-sibling family cohort | |
|---|---|---|---|---|---|---|
| | Number of cases (crude incidence rate, per 1000 person years), exposed twins/ unexposed twins | HR (95% CI)* | Number of cases (crude incidence rate, per 1000 person years), exposed twins /matched singletons | HR (95% CI) † | Number of cases (crude incidence rate, per 1000 person years), exposed twins/full siblings | HR (95% CI) £ |
| By gender of the surviving twins | | | | | | |
| Male | 103 (13.08)/334 (7.87) | 1.74 (1.37–2.22) | 103 (13.08)/366 (8.61) | 1.63 (1.29–2.07) | 76 (12.96)/64 (7.51) | 2.46 (0.77–7.91) |
| Female | 75 (10.94)/266 (7.61) | 1.37 (1.04–1.81) | 75 (10.94)/357 (10.23) | 1.21 (0.93–1.58) | 54 (11.53)/66 (8.93) | 1.22 (0.25–5.85) |
| By family history of psychiatric disorders | | | | | | |
| Yes | 20 (17.98)/80 (15.63) | 3.61 (0.43–30.1) | 20 (17.98)/106 (19.17) | 0.79 (0.25–2.56) | 12 (18.00)/16 (12.32) | - |
| No | 158 (11.60)/520 (7.20) | 1.62 (1.33–1.97) | 158 (11.60)/617 (8.58) | 1.44 (1.19–1.74) | 118 (11.94)/114 (7.80) | 1.55 (0.80–3.00) |
| By gender difference of the twin pair | | | | | | |
| Same-sex twin pair | 130 (12.25)/423 (7.56) | 1.69 (1.34–2.12) | 130 (12.25)/78 (7.97) | 1.78 (1.21–2.63) | 98 (12.70)/93 (7.90) | 1.50 (0.73–3.11) |
| Opposite-sex twin pair | 48 (11.65)/177 (8.27) | 1.30 (0.81–2.10) | 48 (11.65)/38 (9.64) | 1.18 (0.61–2.28) | 32 (11.29)/37 (8.94) | 0.96 (0.29–3.20) |
| By survival days of the deceased twin | | | | | | |
| 0–6 days | 124 (11.72)/415 (7.58) | 1.57 (1.26–1.94) | 124 (11.72)/516 (9.42) | 1.34 (1.08–1.65) | 91 (12.09)/82 (7.54) | 1.67 (0.70–3.99) |
| 7–27 days | 35 (13.09)/134 (9.21) | 1.35 (0.89–2.05) | 35 (13.09)/142 (9.69) | 1.59 (1.07–2.36) | 26 (13.07)/24 (7.65) | 7.04 (0.79–62.4) |
| 28–59 days | 19 (12.87)/51 (6.33) | 2.56 (1.36–4.81) | 19 (12.87)/65 (8.13) | 1.66 (0.92–2.99) | 13 (12.58)/24 (12.71) | 0.23 (0.03–1.80) |

* Cox regression models were stratified by matching identifiers (sex, birth year, and gestational age), and adjusted for birth weight for gestational age, maternal age at childbirth, low Apgar score (≤7) at 5/10 min, maternal educational level at childbirth, maternal cohabitation status during pregnancy, and family history of psychiatric disorders.

† Cox regression models were stratified by matching identifiers (sex, birth year, gestational age, birth weight for gestational age, birth order), and adjusted for maternal age at childbirth, low Apgar score (≤7) at 5/10 min, maternal educational level at childbirth, maternal cohabitation status during pregnancy, and family history of psychiatric disorders.

£ Cox regression models were stratified by family identifiers, and adjusted for sex, birth year, gestational age, birth weight for gestational age, low Apgar score (≤7) at 5/10 min, maternal educational level at childbirth, and maternal cohabitation status during pregnancy.

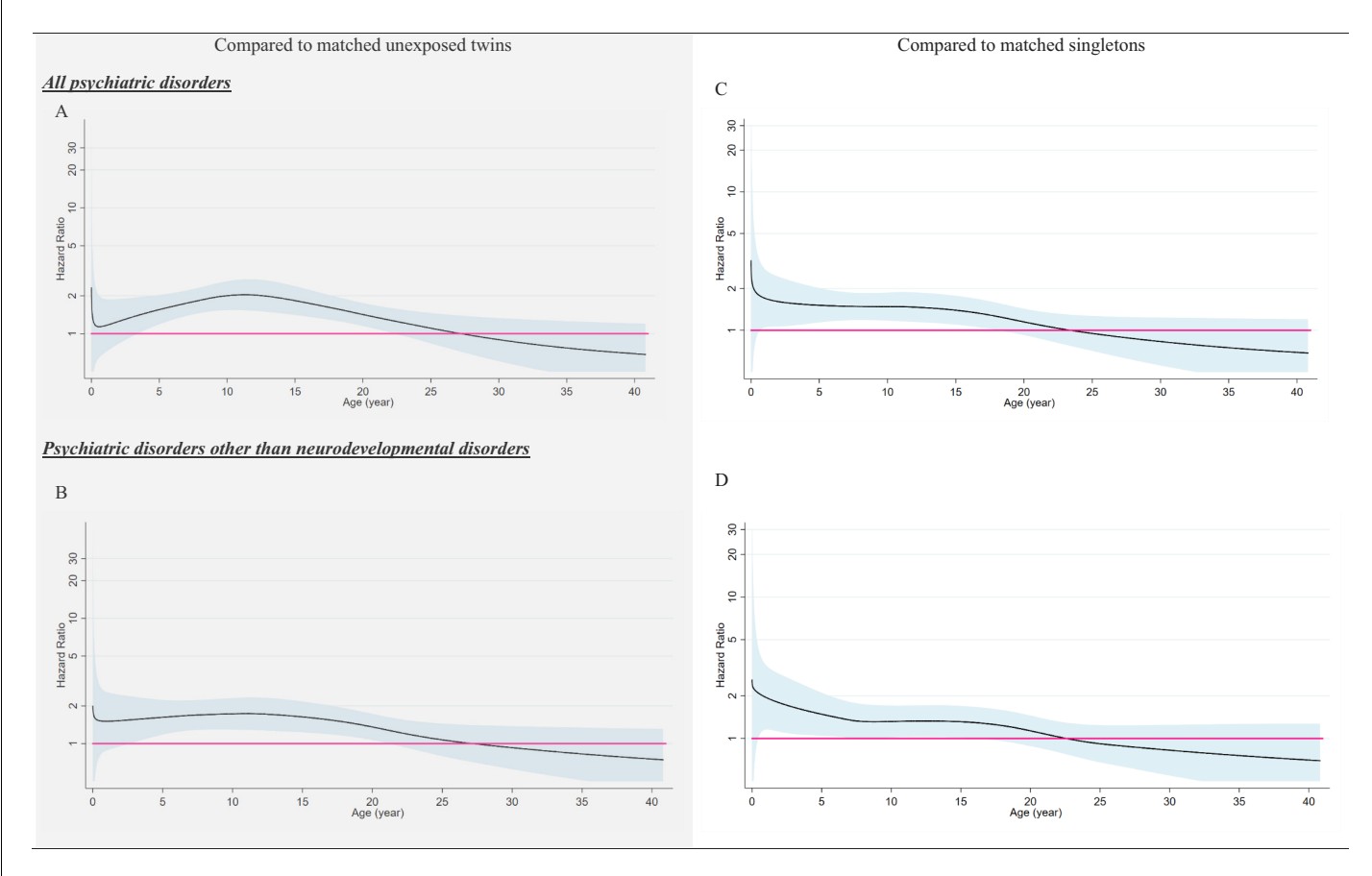

**Figure 2.** The association between loss of a co-twin at birth and subsequent risk of psychiatric disorders by attained age, analyses of population-based matched cohort. [*]Time-varying hazard ratios were derived from flexible parametric survival models, allowing relative risk of psychiatric disorders to vary over attained age. A spline with five df (four intermediate knots and two knots at each boundary, placed at quintiles of distribution of events) was used for the baseline rate, while three df was used for the time-varying effect. All models were adjusted for birth year, sex, gestational age, birth weight for gestational age, birth order, maternal age at childbirth, low Apgar score ($\leq$7) at 5/10 min, maternal educational level at childbirth, maternal cohabitation status during pregnancy, and family history of psychiatric disorders.

The online version of this article includes the following source data for figure 2:

**Source data 1.** Summary data for *Figure 2*.

siblings of the exposed twins, our assessment demonstrates a robust association between early loss of a co-twin and subsequent risk of psychiatric disorders. Given the scarcity of existing data within this area of research, our findings call for further investigation on the possible underlying mechanisms linking the experience of a co-twin loss at birth to mental health decline during adulthood. Particularly, despite the lack of information on zygosity, the higher relative risk observed after early loss of a same-sex co-twin, compared with a loss of opposite-sex co-twin, may indicate the importance of shared genetic background on the formation of a twinship bond. This is similar to the greater grief intensity reported among monozygotic twins who experienced an adult loss of co-twin, compared with dizygotic twins, and consistent with the evolutionary theory suggesting a role of genetic relatedness in the bereavement process (*Segal and Blozis, 2002*; *Segal, 2019*).

The major concern in studies of this kind is that, the death of a co-twin at birth may be an indicator of a poor pregnancy or birth conditions or congenital defects that are shared within a twin pair, and hence vulnerability of the surviving twin to various diseases (such as developmental defects and other somatic disorders) during their later lives. In present analysis, we indeed attempted to have a control for birth characteristics, yet we cannot rule out the possibility that both physical vulnerabilities of the surviving twins and their close contacts with health care due to these somatic problems have contributed to a detection of psychiatric disorders in this population. However, similar results

were obtained in our sensitivity analyses where the presence of congenital abnormalities and the diagnosis of severe somatic conditions during follow-up were taken into account by subgroup analyses or by additional adjustments, suggesting a minor influence of these factors. Another possible explanation for the observed increased risk of psychiatric disorders among the surviving twins could be altered parenting of the grieving parents. Indeed, given that we also observed a heightened risk of psychiatric disorder in full siblings, especially younger full siblings, of the bereaved twins in our twin-sibling family analysis, altered parenting style among bereaved individuals and its impact on offspring's mental health need further investigation. Our additional analyses taking into account clinically confirmed psychiatric disorders of the bereaved parents during the follow-up suggest limited mediating role of clinically confirmed parental psychiatric disorders in the association between early co-twin loss and risk of psychiatric disorder. Nevertheless, severe mental illness requiring a clinical diagnosis affects a relatively small proportion of the bereaved parents.

The major merit of our study is the use of population-based cohort design, including 787 exposed twins with a complete follow-up of up to 41 years. In addition to a full consideration of birth factors, the possible influence of twin pregnancy and twin birth was controlled through the twin-twin comparison, while the social conditions during childhood was taken into consideration in the twin-singleton comparison. Through the twin-full sibling comparison, we were further able to control for potential familial confounders, as well as explore possible underlying mechanisms related to parenting. As the largest cohort study on co-twin loss at birth to date, we had sufficient statistical power to perform most of the planned subgroup analyses. Information bias was minimized since the registration and diagnosis of exposure and outcome was compiled prospectively and independently. Furthermore, the availability of rich information on sociodemographic and medical conditions for both the study participants and their parents enabled considerations of a wide range of important confounding factors.

Limitations include the late establishment of outpatient care records in the Swedish National Patient Register (2001-), which may have rendered an underestimated number of psychiatric disorder diagnoses, especially the milder ones. In addition, individuals were relatively young at the end of follow-up (up to 41 years, with median age at follow-up as 19 years). Therefore, the study focused mainly on early-onset psychiatric disorders. The association between loss of a co-twin at birth and psychiatric disorders beyond early adulthood needs to be addressed in further studies. Furthermore, because we identified individuals through the unique personal identification numbers assigned at birth to each Swedish-born person, our study involved merely twins who lost their co-twin at or shortly after delivery. The loss of a co-twin during the early pregnancy period (i.e., vanishing twin) or due to stillbirth, which might also have psychological effects on the surviving twin (*A Silent Cry, 2008*), is therefore beyond the scope of the present study. Finally, although we made every effort to control for important confounders such as birth characteristics, social and familial conditions, and shared genetic background, we cannot exclude the possibility of residual confounding.

In conclusion, in the Swedish population, exposure to death of a co-twin at birth was associated with a subsequently elevated risk of psychiatric disorders among the surviving twin. These findings call for medical and scientific attention of the mental health of this bereaved population and further exploration of the underlying mechanisms.

## Materials and methods

### Study design

Based on the Swedish Total Population Register, we identified all individuals born in Sweden between 1973 and 2011. Utilizing the personal identification numbers that are uniquely assigned to all Swedish residents, we linked the data to the Multi-Generation Register and identified all twin pairs (i.e., having the same biological father and mother and born on the same day (+/- 1 day), excluding multiple births) and singletons (i.e., the only baby during a single delivery). We obtained prospective information about the study population through cross-linkage to the Medical Birth Register (MBR), the Causes of Death Register, and the National Patient Register (NPR). In order to enter the study, all included individuals had to have survived at least 60 days after birth and have available information in MBR, which has collected nationwide information on deliveries in Sweden since 1973.

## Population-based matched cohort

Because twins have considerably elevated mortality rate during the first and second months after birth and to maximize the sample size of the exposed twins in our study, we defined loss of a co-twin at birth as a death of the co-twin within 60 days after birth, according to information obtained from the Causes of Death Register, which is available electronically for register-based research since 1952. As shown in *Figure 1*, among 88,636 eligible twins, 787 had a co-twin that died at birth or within 60th days after birth and were included in the exposed group. The 60th day after birth was considered as the date of cohort entry (index date) for exposed twins.

We included two reference groups. First, with the aim of controlling for twin pregnancy and twin birth, five unexposed twins (with a co-twin that survived at least 60 days after birth) per exposed twin were randomly selected from the twin population on the index date of the exposed twin (i.e., also the index date for unexposed twins). They were individually matched to the exposed twin by birth year, sex, and GA (<28, 28–31, 32–36, or ≥37 weeks). Second, as the exposed twins grew up on their own, as singletons, in order to control for such social conditions, five singletons per exposed twin were also selected from the singleton population, on the index date of the exposed twin (i.e., also the index date for matched singletons). Because we had a larger pool for selection of matched singletons, compared with selection of unexposed twins, we were able to use more matching variables, including sex, birth year, GA (<28, 28–31, 32–36, or ≥37 weeks), birth weight for GA (<10th, 10–30th, 31–50th, 51–70th, 71–90th, or >90th percentile), and birth order (1st, 2nd, 3rd, ≥4th child within a family), to ensure the comparability between twins and singletons with regard to birth characteristics.

## Twin-sibling family cohort

To address the concern about familial confounders, such as genetic background and environmental factors during childhood shared within a family, we further constructed a twin-sibling family cohort where non-twin full siblings of the exposed twins, if any, identified through the Multi-Generation Register, were included on the index date of the exposed twins or 60 days after their own birth, if born later than the twins (i.e., the index date for non-twin full siblings).

## Follow-up

Follow-up of all study participants started from the index date and lasted until the occurrence of any or a specific type of psychiatric disorders, death, emigration, or the end of follow-up (December 31, 2013), whichever occurred first.

## Psychiatric disorders

Any first-ever inpatient or outpatient hospital visit with a psychiatric disorder as one of the registered diagnoses, primary or secondary, during the follow-up was identified from the NPR (ICD eight and ICD nine codes: 290–315, ICD 10 codes: F00-F99). For sub-analyses on the three categories of psychiatric disorders, including neurodevelopmental disorders (i.e., attention deficit hyperactivity disorder, autism spectrum disorder, and intellectual disabilities), emotional disorders (i.e., depression, anxiety, and stress-related disorders), and other psychiatric disorders, first-ever diagnosis of each specific group was also extracted from the NPR, according to corresponding ICD codes shown in *Supplementary file 1*-Table 1.

## Covariables

Information on birth factors of the study participants, including GA, birth weight, and Apgar score, as well as maternal characteristics, including maternal age, educational level, and cohabitation status at childbirth, was extracted from the MBR. Specifically, to calculate the birth weight for gestational age percentiles, we generated birth weight z-score for GA, sex, and twin/singleton-specific populations and categorized it into <10th, 10–30th, 31–50th, 51–70th, 71–90th, and >90th percentile. We considered a poor neonatal condition as the presence of low Apgar score (≤7) at 5 or 10 min after birth. Family history of psychiatric disorders was defined as any diagnosis of or death due to psychiatric disorder or suicide among the first-degree relatives (i.e., biological parents and siblings) of the study participants, according to the NPR or Causes of Death Register. In sensitivity analyses, to detect the potential role of declined mental health among the bereaved parents on the association

of interest, we extracted diagnoses of psychiatric disorders among parents of study participants during follow-up from the NPR. Furthermore, to take into consideration the possible physical weakness of the surviving twins compared with their reference individuals, we collected data on the presence of congenital abnormalities and severe somatic diseases during follow-up from the NPR (see *Supplementary file 1*-Table 1). The study was approved by the Regional Ethics Review Board in Stockholm, Sweden.

## Statistical analysis

We estimated the association between loss of a co-twin at birth and risk of psychiatric disorders using HRs with 95% CIs, derived from conditional Cox regression models where the attained age was applied as the underlying time scale.

In the population-based matched cohort, exposed twins were compared to matched unexposed twins and matched singletons. In addition to the matching variables, the models were further adjusted for birth weight for GA (<10th, 10–30th, 31–50th, 51–70th, 71–90th, >90th, or unknown, for twin-twin comparison only), maternal age at childbirth (<28, 29–32, or $\geq$33 years), low Apgar score ($\leq$7) at 5/10 min (yes, no, or unknown), maternal educational level at childbirth (<9, 9–12, >12 years, or unknown), maternal cohabitation status during pregnancy (non-cohabitation, cohabitation, or unknown), and family history of psychiatric disorders (yes or no). In subgroup analyses, we calculated the HRs by sex (male or female), sex difference of the twin pair (same-sex or opposite-sex), and family history of psychiatric disorders (yes or no). We also subgrouped exposed twins by survival days of the deceased twins (0–6, 7–27, or 28–59 days). In addition to considering all psychiatric disorders as one group, we did sub-analyses for three categories of psychiatric disorders, that is neurodevelopmental disorders, emotional disorders, and other psychiatric disorders. We further visualized the change of HR by attained age using flexible parametric models.

Next, we repeated the main analyses in the twin-sibling family cohort. Cox models were stratified by family identifiers, and adjusted for birth year, sex, GA, as well as all covariables used in the population-based matched cohort. In addition to a comparison between the exposed twins and all their full siblings, we also compared the exposed twins with their older or younger full siblings, separately.

In sensitivity analyses, we explored the effect of mental health of parents on the observed association by considering the occurrence of psychiatric disorders among parents diagnosed during the follow-up (yes or no) in a subgroup analysis. In addition, to alleviate concerns that the observed associations were mainly attributed to the suboptimal somatic conditions of the surviving twins than their matched individuals, we did subgroup analyses by, or additionally adjusted for, the presence of congenital abnormalities or the diagnosis of severe somatic conditions during follow-up. Lastly, to test the robustness of the observed association to the definition of 'death at birth,' we re-ran the analyses by using 28 days (i.e., neonatal death), instead of 60 days, after birth for the definition of loss of a co-twin at birth. All analyses were conducted in SAS statistical software, version 9.4 (Cary, NC) and STATA 15 (StataCorp LP).

## Additional information

### Funding

| Funder | Grant reference number | Author |
| --- | --- | --- |
| Icelandic Centre for Research | 163362-051 | Unnur A Valdimarsdóttir |
| European Research Council | 726413 | Unnur A Valdimarsdóttir |
| Swedish Research Council | 340-2013-5867 | Catarina Almqvist |
| National Science Foundation | 81971262 | Huan Song |

The funders had no role in study design, data collection and interpretation, or the decision to submit the work for publication.

## Author contributions

Huan Song, Conceptualization, Formal analysis, Funding acquisition, Methodology, Writing - original draft, Writing - review and editing; Fang Fang, Conceptualization, Supervision, Methodology, Writing - original draft, Writing - review and editing; Henrik Larsson, Data curation, Supervision, Methodology, Writing - review and editing; Nancy L Pedersen, Supervision, Methodology, Writing - review and editing; Patrik KE Magnusson, Data curation, Methodology, Writing - review and editing; Catarina Almqvist, Funding acquisition, Methodology, Writing - review and editing; Unnur A Valdimarsdóttir, Conceptualization, Supervision, Funding acquisition, Methodology, Writing - original draft, Writing - review and editing

## Author ORCIDs

Huan Song (iD) https://orcid.org/0000-0003-3845-8079
Fang Fang (iD) http://orcid.org/0000-0002-3310-6456
Unnur A Valdimarsdóttir (iD) https://orcid.org/0000-0001-5382-946X

## Ethics

Human subjects: The study was approved by the Regional Ethics Review Board in Stockholm, Sweden (Dnr 2013/862-31/5); and the requirement of informed consent was waived for register-based studies in Sweden.

## Decision letter and Author response

Decision letter https://doi.org/10.7554/eLife.63514.sa1
Author response https://doi.org/10.7554/eLife.63514.sa2

# Additional files

## Supplementary files

• Source code 1. SAS script for the primary analyses.

• Supplementary file 1. Supplementary tables. Supplementary Table 1. International Classification of Diseases (ICD), eighth (ICD-8; 1969–1986), ninth (ICD-9; 1987–1996), and tenth (ICD-10; 1997–2013) revisions codes for diagnoses used in this study Supplementary Table 2. Hazard ratios (HRs) with 95% confidence intervals (CIs) for any psychiatric disorder among twins who lose a co-twin at birth, calculated separately for older and younger siblings of the surviving twins, subgroups by psychiatric disorders among parents during follow-up Supplementary Table 3. Hazard ratios (HRs) with 95% confidence intervals (CIs) for any psychiatric disorder among twins who lose a co-twin at birth, subgrouped by or additionally adjusted for diagnosis of congenital abnormalities and severe somatic diseases during follow-up Supplementary Table 4. Hazard ratios (HRs) with 95% confidence intervals (CIs) for any psychiatric disorder among twins who lost a co-twin within 28 days after birth, derived from different Cox models and by subtypes of psychiatric disorders

• Transparent reporting form

## Data availability

One source data file has been provided. Original data are held by Swedish National Board of Health and Welfare, Statistics Sweden and the Swedish Twin Registry. Due to Swedish law on data protection and the ethical approval of the current study, we cannot make the data publicly available. However, any researcher can access the data by obtaining an ethical approval from a regional ethical review board and thereafter request the original data from the Swedish National Board of Health and Welfare, Statistics Sweden, and the Swedish Twin Register. Detailed information on data application can be found at https://www.registerforskning.se/en/ and https://ki.se/en/research/the-swedish-twin-registry.

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
