## [Decision Letter]

**Acceptance summary:**

Song and colleagues estimate the risk of psychiatric disorders in the surviving twin following the death of a co-twin at birth. The study leverages the excellent Swedish registry system, it includes informative sample groups that are large in size and examines a considerable number of background characteristics that have possible associations with psychiatric symptoms and disorders in surviving twins. The major finding was an excess of psychiatric disorders in twins who co-twins died at a very early age, relative to twins from intact twin pairs and full siblings. Overall, the study is a valuable addition to the literature on the mental health of bereaved twins.

**Decision letter after peer review:**

Thank you for submitting your article "Loss of a co-twin at birth and subsequent risk of psychiatric disorders" for consideration by *eLife*. Your article has been reviewed by three peer reviewers, including Maria H Chahrour as the Reviewing Editor and Reviewer #1, and the evaluation has been overseen by Eduardo Franco as the Senior Editor.

The reviewers have discussed the reviews with one another and the Reviewing Editor has drafted this decision to help you prepare a revised submission.

Summary:

In this manuscript, Song and colleagues estimate the risk of psychiatric disorders in the surviving twin following the death of a co-twin at birth. The same group had published a study in *eLife* earlier this year, reporting an increased risk of psychiatric disorders in twins who had lost a co-twin during childhood or adulthood. In the current study, the group analyzed data from a Swedish cohort of 787 twins who lost their co-twins at birth (defined within 60 days from birth). They compared the rate of psychiatric disorders in these twins to 3,935 matched unexposed twins, 3,935 matched unexposed singletons, and 880 full sibs of the exposed twins. They found that the exposed twins were at an increased risk of developing psychiatric disorders compared to unexposed twins with an HR=1.56. Compared to matched singletons, the HR was 1.41. The study leverages the excellent Swedish registry system, it includes informative sample groups that are large in size and examines a considerable number of background characteristics (e.g., gestational age, birth weight) that have possible associations with psychiatric symptoms and disorders in surviving twins. The major finding was an excess of psychiatric disorders in twins who co-twins died at a very early age, relative to twins from intact twin pairs and full siblings. Overall, the study is a valuable addition to the literature on the mental health of bereaved twins. The strength of the study is the population-based cohort design with a 41-year follow-up and the consideration of multiple confounders in their analyses. However, there are several issues that need to be addressed.

Essential revisions:

1) It was not clear to me how the authors controlled for social conditions and familial factors? Especially the influence of bereaved parents on the mental health and well-being of the surviving twin. Since this is a measure variable, please elaborate on this in the Materials and methods.

2) The authors state that: "Nevertheless, our further analyses by clinically confirmed psychiatric disorders of the bereaved parents during the follow-up found limited effect of this factor on the observed associations."

The bereaved parents do not necessarily need to have a formally diagnosed psychiatric disorder in order to influence the mental health of the exposed surviving twin. The authors should acknowledge this in the Discussion and discuss the influence of bereaved parents on the mental health and well-being of the surviving twin as a confounding factor.

3) The part of the Discussion which starts: "In addition to the genetic relatedness" and ends with "genetic background on the formation of such a twinship bond." is inappropriate and out of place. The whole section should be removed. It's based on speculations, not scientific data, and does not constitute a discussion of the results or data presented in this manuscript.

4) Please clarify this sentence in the Discussion "Finally, although we made every effort to control for all relevant confounders, we cannot exclude the possibility that residual confounding."

5) One of the main concerns of this study is the similarity of the current manuscript to a recently published work in the same journal (Song et al., *eLife*). The authors have failed to convince the reviewers that this additional complementary analysis is innovative enough to contribute to a manuscript; what additional information does this paper bring to the literature? Also, the rationale leading to the research question in not well articulated, i.e., the knowledge gap leading to the research question (which is also not well defined; see our comments below) needs to be clearly stated.

6) The authors stated that they want "to explore a potential rise in rate of psychiatric disorders among surviving twins after loss of a co-twin at birth". Ideally, when writing the aim of epidemiological analytical studies, one should state the direction of the association they want to test as well as to use verbs that somehow represent the proposed analytical strategy. For example, using verbs such as “explore” is vague and may give the sense of fishing, that is, looking for statistically significant results. This type of verbs is more appropriate for qualitative research. One way to rephrase this aim would be something along the lines of “to estimate the extent to which loss of a co-twin at birth is associated with rate of psychiatric disorders among surviving twins.”

7) Another issue that needs to be addressed is the definition of exposed twins, which is those who lost a co-twin within 60 days after birth. This long timespan might bias the results; an exposed twin having lost their co-twin immediately at birth might be substantially different than an exposed twin having lost their co-twin 60 days after birth. This is because the time span of 60 days might allow emotional bonds to form between twins. The authors should justify their choice of 60 days after birth as a cut off. Similarly, exposure to a grieving parent during the first 60 years of life may have a confounding effect. Although this was accounted for by recruitment of the twins' full siblings as control group, exposure of full siblings to parental grief may not necessarily impose the same risk as in the surviving twins. This is because the siblings might be in different developmental stages with different needs and dependency levels. Moreover, the manuscript would benefit from an explanation on how missing data were managed.

8) The authors could add the clinical implications of the study to the Discussion section to improve the strength of their arguments. For instance, this study can inform future interventions to support the surviving twins to alleviate the mental health impact of the loss of a co-twin.

9) There is no mention of zygosity, a variable that is understandably missing from the sample except in the case of opposite-sex twins. It is possible that surviving MZ twins would feel the loss more strongly than surviving DZ twins, mirroring the relationships we see in the adult twin population. This issue should at least be addressed. Of course, twine would only know their zygosity if they learned it from a reliable medical record.

10) There are relevant books and research papers that have not been referenced: Woodward, 2010; Woodward, 1988; Segal, 2019.

11) There is a passage in the author's paper suggesting that twins bond in the womb;

“It has indeed been proposed that twins actually begin their co-twin identity formation in the womb, so called “in-utero bond” (21). This notion gains support by the fact that twins share cellular origins and the womb environment during the fetal development period, and the interactive patterns of behaviors in womb have been observed and documented by researchers using ultrasound (22, 23).”

The presence in this passage in the present paper is unfortunate because it only supports a baseless and romantic notion. This must be rephrased or omitted. Many reared-apart twins have no awareness of being a twin. Many of these reared-apart twins are adoptees and their feelings of loneliness or emptiness are better explained by lack of resemblance between themselves and their family members-many non-twin adoptees also express such feelings.

See this from a recent book, Twin Mythconceptions:

Prenatal twins' interactive behaviors do not appear to be expressed with any intention or awareness of the other. Low oxygen tension in fetal blood, as well as pregnanolone and prostaglandin D2 that are provided by the placenta, keep the fetus sedated [19]. If prenatal cotwins' interactive activities influence the nature of their postnatal relationship, then identical twins should show more sustained coordinated behaviors in the womb than fraternal twins, but that is not the case. Research conducted in 2012 found no evidence that fetal dichorionic twins' body movements and rest-sleep cycles are coordinated, challenging some previous reports. It seems, instead, that any synchronized behaviors displayed by twins are infrequent, brief, and unintentional [20].

12) In my experience, some parents report that young singleton twins seem to crave physical contact. Perhaps they miss the tactile sensations that the company of the other twin provides, but that is also highly speculative. It seems more likely that surviving twins' increased psychiatric disorders may be linked to parenting issues (e.g., overprotection), even when twins are not told they had a twin, or to whatever physical factor was responsible for the demise of the co-twin. Finally, we sometimes hear stories from therapists and adults that current depression of some clients is linked to a lost twin, but there is no evidence that the person was a twin. I believe that, as scientists, the authors of this paper would be likely to dismiss such stories; to link depression with loss of a twin at birth or soon after is irresponsible on the part of therapists. At best we can say that more work in this area needs to be done.

---

## [Author Response]

Essential revisions:1) It was not clear to me how the authors controlled for social conditions and familial factors? Especially the influence of bereaved parents on the mental health and well-being of the surviving twin. Since this is a measure variable, please elaborate on this in the Materials and methods.

Thank you for your comments. We did not have a lot of information on social conditions and familial factors because of the register-based nature of the study. But we did consider several main characteristics, including maternal age, educational level, and cohabitation status at childbirth as well as birth order of the index child. Furthermore, the control of social conditions and familial factors was mainly achieved by contrasting the exposed twins to a reference group of individuals with reasonably comparable conditions on these factors. For instance, one distinct social condition of a twin life is that they grow up with the close company of their twin partner. This is however not the case for twins who lost their co-twin at birth—they grow up like singletons. Therefore, in addition to having a reference group of matched twins who didn’t experience a twin loss at birth (i.e., the twin-twin comparison) to control for twin pregnancy and twin birth, we also compared the exposed twins with a group of singletons matched by age, sex, and birth characteristics (i.e., the twin-singleton comparison).

Similarly, we did not have detailed information on various familial factors. However, the family-related conditions, as a whole, were taken into consideration through the comparison between exposed twins and their full siblings (i.e., the twin-sibling family cohort), since it’s reasonable to assume that, to a large extent, full siblings (i.e., having the same biological father and mother) should have been equally influenced by factors that cluster within a family, such as genetic background and environmental factors during childhood (including the parenting of the bereaved parents). To clarify this further, we highlighted the aims, as well as the controlled factors, in each planned comparison, in the revised manuscript.

Results section:

“The first reference group included 3,935 sex-, birth year-, and gestational age (GA)-matched unexposed twins randomly selected from the Swedish twin population that did not experience such a loss, to control for twin pregnancy and twin birth. Because the exposed twins grew up on their own, as singletons, which differs from the social conditions of a twin life, we also included 3,935 singletons randomly selected from the singleton population that were individually matched to the exposed twins on sex, birth year, and birth factors (GA, birth weight for GA, and birth order) as the second reference group. In addition to the population-based matched cohort, to address the concern of familial factors such as genetic background and environmental factors during childhood shared within a family (e.g., parenting of the bereaved parents), we compared 569 exposed twins to their full siblings (n=880) in twin-sibling family cohort.”

Discussion section:

“Notably, this association was independent of multiple important confounders, including birth characteristics, childhood social conditions (by comparing bereaved twins to singletons), and other familial factors (by comparing bereaved twins to their full siblings), indicating that increased clinical alertness of the mental health of surviving twins after a very early co-twin loss is warranted.”

Materials and methods section:

“We included two reference groups. First, with the aim of controlling for twin pregnancy and twin birth, five unexposed twins (with a co-twin that survived at least 60 days after birth) per exposed twin, were randomly selected from the twin population on the index date of the exposed twin (i.e., also the index date for unexposed twins). They were individually matched to the exposed twin by birth year, sex, and GA (<28, 28-31, 32-36, or ≥37 weeks). Second, as the exposed twins grew up on their own, as singletons, in order to control for such social conditions, five singletons per exposed twin were also selected from the singleton population, on the index date of the exposed twin (i.e., also the index date for matched singletons).”

Materials and methods section:

“To address the concern about familial confounders, such as genetic background and environmental factors during childhood shared within a family, we further constructed a twin-sibling family cohort where non-twin full siblings of the exposed twins….”

2) The authors state that: "Nevertheless, our further analyses by clinically confirmed psychiatric disorders of the bereaved parents during the follow-up found limited effect of this factor on the observed associations."The bereaved parents do not necessarily need to have a formally diagnosed psychiatric disorder in order to influence the mental health of the exposed surviving twin. The authors should acknowledge this in the Discussion and discuss the influence of bereaved parents on the mental health and well-being of the surviving twin as a confounding factor.

We agree with the reviewer that by identifying the clinical diagnosis of psychiatric disorders among bereaved parents alone, we cannot fully address the influence of altered parenting by the bereaved parents on the studied association, as the parents do not need to have mental problems to give unfavorable parenting.

However, we consider the altered parenting to be more likely a mediator, rather than a confounder, in the studied association. Namely, it is possible that altered parenting style resulting from the bereavement may be one of the reasons why surviving twins suffer increased risk of psychiatric disorders after early co-twin loss. In the revised manuscript, we have now extended a discussion on this in the Discussion.

Discussion section:

“Another possible explanation for the observed increased risk of psychiatric disorders among the surviving twins could be altered parenting of the grieving parents. Indeed, given that we also observed a heightened risk of psychiatric disorder in full siblings, especially younger full siblings, of the bereaved twins in our twin-sibling family analysis, altered parenting style among bereaved individuals and its impact on offspring’s mental health needs further investigation. Our additional analyses taking into consideration clinically confirmed psychiatric disorders of the bereaved parents during the follow-up suggest limited mediating role of clinically diagnosed parental psychiatric disorders in the association between early co-twin loss and risk of psychiatric disorder. Nevertheless, severe mental illness requiring a clinical diagnosis affects a relatively small proportion of the bereaved parents.”

3) The part of the Discussion which starts: "In addition to the genetic relatedness" and ends with "genetic background on the formation of such a twinship bond." is inappropriate and out of place. The whole section should be removed. It's based on speculations, not scientific data, and does not constitute a discussion of the results or data presented in this manuscript.

Thank you for this important comment. We agree and have now removed these speculative statements.

4) Please clarify this sentence in the Discussion "Finally, although we made every effort to control for all relevant confounders, we cannot exclude the possibility that residual confounding."

We have clarified the sentence as below:

Discussion section:

“Finally, although we made every effort to control for important confounders such as birth characteristics, social and familial conditions, and shared genetic background, we cannot exclude the possibility of residual confounding.”

5) One of the main concerns of this study is the similarity of the current manuscript to a recently published work in the same journal (Song et al., 2020). The authors have failed to convince the reviewers that this additional complementary analysis is innovative enough to contribute to a manuscript; what additional information does this paper bring to the literature? Also, the rationale leading to the research question in not well articulated, i.e., the knowledge gap leading to the research question (which is also not well defined; see our comments below) needs to be clearly stated.

Thank you for your comments. We believe that the present study adds important new knowledge to our previous work. Below are our two main arguments.

First, although bereavement due to a co-twin loss and its impact on risk of psychiatric disorders has been reported among adult twins, including our own work (as the reviewer pointed out), there is a clear lack of knowledge concerning the potential impact of exposure to a co-twin death at birth (before any memory of the deceased twin can be consolidated) on the risk of psychiatric disorders among the surviving twins.

Second, the absence of such knowledge is mainly due to the complexity of the research question and scarcity of rigorous, informative data. In addition to the unique co-twin loss experience at birth, the surviving twins may be exposed to many other factors, such as the possible genetic vulnerability to diseases, suboptimal birth characteristics (due to the twin birth), a singleton-like life, and living in a family with experience of bereavement, etc. which might all influence their risk of future psychiatric disorders. Therefore, with the Swedish nationwide data containing more than 800 exposed twins and detailed information on birth characteristics and family structure, we have a unique opportunity to address this question.

As most of co-twin loss occur at an older age (median age at loss=59 year), the majority of the study population in our previous study was born before 1973 and therefore had no detailed information from the Medical Birth Register. We therefore could not appropriately address and discuss such a question in our last *eLife* paper.

In the revised manuscript, we have now emphasized the motivation and novelty of the present study, in both the Introduction and Discussion.

Introduction section:

“In the total absence of data on the rate of psychiatric disorders among twins who lost a co-twin at birth, we conducted a nationwide population- and sibling-matched cohort study to estimate the extent to which loss of a co-twin at birth is associated with the incidence of psychiatric disorders among surviving twins, after carefully controlling for important confounders such as birth characteristics and familial factors.”

Discussion section:

“Notably, this association was independent of multiple important confounders, including birth characteristics, childhood social conditions (by comparing bereaved twins to singletons), and other familial factors (by comparing bereaved twins to their full siblings), indicating that increased clinical alertness of the mental health of surviving twins after a very early co-twin loss is warranted.”

Discussion section:

“While accumulating evidence supports that both childhood and adult twin loss are associated with increased risk of psychiatric morbidity among the surviving twins (10, 11), no previous study has addressed whether such emotional reactions can be observed after a very early co-twin loss where limited twin relationship, perception or memory from the loss could be expected. The absence of evidence is mainly due to the complexity of the research question and lack of high-quality data to address potential confounding by multiple factors, such as twin pregnancy and birth (i.e., suboptimal birth characteristics) but a singleton-like life, familial factors, and genetic susceptibility to diseases. Therefore, with the unique Swedish nationwide data sources, which provide a substantial sample size of exposed twins with detailed data on birth characteristics and familial information, we conducted the present study. By contrasting the rate of psychiatric disorders among surviving twins who were exposed to a co-twin loss at birth with that of several comparison groups, including matched unexposed twins and singletons as well as the full siblings of the exposed twins, our assessment demonstrates a robust association between early loss of a co-twin and subsequent risk of psychiatric disorders.”

6) The authors stated that they want "to explore a potential rise in rate of psychiatric disorders among surviving twins after loss of a co-twin at birth". Ideally, when writing the aim of epidemiological analytical studies, one should state the direction of the association they want to test as well as to use verbs that somehow represent the proposed analytical strategy. For example, using verbs such as “explore” is vague and may give the sense of fishing, that is, looking for statistically significant results. This type of verbs is more appropriate for qualitative research. One way to rephrase this aim would be something along the lines of “to estimate the extent to which loss of a co-twin at birth is associated with rate of psychiatric disorders among surviving twins.”

Thank you for the comment. The request changes have been made. Please see the new statement below:

Introduction section:

“In the total absence of data on the rate of psychiatric disorders among twins who lost a co-twin at birth, we conducted a nationwide population- and sibling-matched cohort study to estimate the extent to which loss of a co-twin at birth is associated with the incidence of psychiatric disorders among surviving twins, after carefully controlling for important confounders such as birth characteristics and familial factors.”

7) Another issue that needs to be addressed is the definition of exposed twins, which is those who lost a co-twin within 60 days after birth. This long timespan might bias the results; an exposed twin having lost their co-twin immediately at birth might be substantially different than an exposed twin having lost their co-twin 60 days after birth. This is because the time span of 60 days might allow emotional bonds to form between twins. The authors should justify their choice of 60 days after birth as a cut off. Similarly, exposure to a grieving parent during the first 60 years of life may have a confounding effect. Although this was accounted for by recruitment of the twins' full siblings as control group, exposure of full siblings to parental grief may not necessarily impose the same risk as in the surviving twins. This is because the siblings might be in different developmental stages with different needs and dependency levels. Moreover, the manuscript would benefit from an explanation on how missing data were managed.

Thank you for this important comment. We agree with the reviewer that the choice of “60 days after birth” needs justifications. Neonatal death is usually defined within 28 days after birth. We used slightly longer time window in the study, because: 1) there was a significantly higher mortality during 28-60 days after birth, compared to thereafter, among twins (e.g., twice as high mortality compared to the month after) and 2) by extending with 28 days to 60 days after birth, we increased the sample size of the exposed twins by 10%. We have now clarified this in the revised manuscript:

Materials and methods section:

“Because twins have considerably elevated mortality rate during the first and second months after birth and to maximize the sample size of the exposed twins in our study, we defined loss of a co-twin at birth as a death of the co-twin within 60 days after birth, according to information obtained from the Causes of Death Register, which is available electronically for register-based research since 1952.”

In addition, to test the impact of different exposure windows on our estimates, we performed subgroup analysis by survival days of the deceased twin (0-6, 7-27, and 28-59 days, see Author response table 1 and Table 3 in the manuscript), which reveals comparable estimates (the statistical power is however limited in the related analyses of twin-sibling family cohort). Also, in another sensitivity analysis, we obtained largely similar results by using 28 days to define the exposed twins (see Author response table 1 and Supplementary table 4 in Supplementary file 1 in the manuscript).

8) The authors could add the clinical implications of the study to the Discussion section to improve the strength of their arguments. For instance, this study can inform future interventions to support the surviving twins to alleviate the mental health impact of the loss of a co-twin.

Thank you for this important comment. The request changes have been made.

Discussion section:

“Notably, this association was independent of multiple important confounders, including birth characteristics, childhood social conditions (by comparing bereaved twins to singletons), and other familial factors (by comparing bereaved twins to their full siblings), indicating that increased clinical alertness of the mental health of surviving twins after a very early co-twin loss is warranted.”

9) There is no mention of zygosity, a variable that is understandably missing from the sample except in the case of opposite-sex twins. It is possible that surviving MZ twins would feel the loss more strongly than surviving DZ twins, mirroring the relationships we see in the adult twin population. This issue should at least be addressed. Of course, twine would only know their zygosity if they learned it from a reliable medical record.

As the reviewer correctly guessed, information on zygosity is not available for all twins apart from the opposite-sex twins. To clarify this, we have added a statement in the Discussion part of the revised manuscript.

Discussion section:

“Particularly, despite of the lack of information on zygosity, the higher relative risk observed after early loss of a same-sex co-twin, compared with a loss of opposite-sex co-twin, may indicate the importance of shared genetic background on the formation of such a twinship bond. This is similar to the greater grief intensity reported among monozygotic twins who experienced an adult loss of co-twin compared with dizygotic twins, and consistent with the evolutionary theory suggesting a role of genetic relatedness in the bereavement process (18,19).”

10) There are relevant books and research papers that have not been referenced: Woodward, 2010; Woodward, 1988; Segal, 2019.

Thank you for the information. We have added these references to the revised manuscript.

Introduction section:

“Previous studies indicate that loss of a co-twin by death in childhood or adulthood is associated with considerable mental morbidities among the surviving twins (Segal, 1993; Woodward, 1988).”

Introduction section:

“However, several scientific and media accounts describe unexpected lingering sorrow among twins who lost their co-twin at or shortly after birth, even among the twins that didn’t know they were born as twins ( Morgan MR 2014; lone twin network: https://lonetwinnetwork.org.uk/members-stories/loss-at-birth/; Woodward J 2010).”

Discussion section:

“This is similar to the greater grief intensity reported among monozygotic twins who experienced an adult loss of co-twin, compared with dizygotic twins, and consistent with the evolutionary theory suggesting a role of genetic relatedness in the bereavement process (Segal, 2002; Segal, 2019).”

11) There is a passage in the author's paper suggesting that twins bond in the womb;“It has indeed been proposed that twins actually begin their co-twin identity formation in the womb, so called “in-utero bond” (21). This notion gains support by the fact that twins share cellular origins and the womb environment during the fetal development period, and the interactive patterns of behaviors in womb have been observed and documented by researchers using ultrasound (22, 23).”The presence in this passage in the present paper is unfortunate because it only supports a baseless and romantic notion. This must be rephrased or omitted. Many reared-apart twins have no awareness of being a twin. Many of these reared-apart twins are adoptees and their feelings of loneliness or emptiness are better explained by lack of resemblance between themselves and their family members-many non-twin adoptees also express such feelings.See this from a recent book, Twin Mythconceptions:Prenatal twins' interactive behaviors do not appear to be expressed with any intention or awareness of the other. Low oxygen tension in fetal blood, as well as pregnanolone and prostaglandin D2 that are provided by the placenta, keep the fetus sedated [19]. If prenatal cotwins' interactive activities influence the nature of their postnatal relationship, then identical twins should show more sustained coordinated behaviors in the womb than fraternal twins, but that is not the case. Research conducted in 2012 found no evidence that fetal dichorionic twins' body movements and rest-sleep cycles are coordinated, challenging some previous reports. It seems, instead, that any synchronized behaviors displayed by twins are infrequent, brief, and unintentional [20].

Thank you for pointing out this important issue. In the revised manuscript, we have tried to avoid the speculation of “in-utero bond”, in line with the comments from the reviewers and editors, by removing all related statements and references.

12) In my experience, some parents report that young singleton twins seem to crave physical contact. Perhaps they miss the tactile sensations that the company of the other twin provides, but that is also highly speculative. It seems more likely that surviving twins' increased psychiatric disorders may be linked to parenting issues (e.g., overprotection), even when twins are not told they had a twin, or to whatever physical factor was responsible for the demise of the co-twin. Finally, we sometimes hear stories from therapists and adults that current depression of some clients is linked to a lost twin, but there is no evidence that the person was a twin. I believe that, as scientists, the authors of this paper would be likely to dismiss such stories; to link depression with loss of a twin at birth or soon after is irresponsible on the part of therapists. At best we can say that more work in this area needs to be done.

Thank you for sharing these experiences and we agree. In the revised manuscript, we emphasize now the currently limited evidence for the link between co-twin loss at birth and future psychiatric disorders and its underlying mechanisms and call for further research in this field.

Discussion section:

“Given the scarcity of existing data within this area of research, our findings call for further investigation on the possible underlying mechanisms linking the experience of a co-twin loss at birth to mental health decline during adulthood.”

Discussion section:

“These findings call for medical and scientific attention of the mental health of this bereaved population and further exploration of the underlying mechanisms.”

**Author response table 1. resptable1:** Hazard ratios (HRs) with 95% confidence intervals (CIs) for any psychiatric disorder among the surviving twins after co-twin loss at birth, by survival days of the deceased twin or using “28 days” to define the exposed twins.

	**Population-based matched cohort**	**Twin-sibling family cohort**				
	Number of cases (Crude incidence rate, per 1000 person years), exposed **twins**/unexposed **twins**	HR(95% CI) ^*^	Number of cases (Crude incidence rate, per 1000 person years), exposed **twins** /matched **singletons**	HR(95% CI) ^†^	Number of cases (Crude incidence rate, per 1000 person years), exposed **twins**/**full siblings**	HR(95% CI) ^£^
***Main analysis***	178(12.08)/600(7.76)	1.56 (1.30-1.87)	178(12.08)/723(9.33)	1.41 (1.19-1.69)	130(12.32)/130(8.17)	1.43 (0.82-2.49)
***Subgroup analysis: by survival days of the deceased twin***						
0-6 days	124(11.72)/415(7.58)	1.57 (1.26-1.94)	124(11.72)/516(9.42)	1.34 (1.08-1.65)	91(12.09)/82(7.54)	1.67 (0.70-3.99)
7-27 days	35(13.09)/134(9.21)	1.35 (0.89-2.05)	35(13.09)/142(9.69)	1.59 (1.07-2.36)	26(13.07)/24(7.65)	7.04 (0.79-62.4)
28-59 days	19(12.87)/51(6.33)	2.56 (1.36-4.81)	19(12.87)/65(8.13)	1.66 (0.92-2.99)	13(12.58)/24(12.71)	0.23 (0.03-1.80)
***Sensitivity analysis: twins who lost a co-twin within 28 days after birth***	159(12.00)/549(7.92)	1.52 (1.25-1.84)	159(12.00)/658(9.47)	1.38 (1.15-1.67)	117(12.29)/106(7.56)	1.74 (0.88-3.45)

^*^ Cox regression models were stratified by matching identifiers (sex, birth year, and gestational age), and adjusted for birth weight for gestational age, maternal age at childbirth, low Apgar score (≤7) at 5/10 min, maternal educational level at childbirth, maternal cohabitation status during pregnancy, and family history of psychiatric disorders.

^†^ Cox regression models were stratified by matching identifiers (sex, birth year, gestational age, birth weight for gestational age, birth order), and adjusted for maternal age at childbirth, low Apgar score (≤7) at 5/10 min, maternal educational level at childbirth, maternal cohabitation status during pregnancy, and family history of psychiatric disorders.

^£^ Cox regression models were stratified by family identifiers, and adjusted for sex, birth year, gestational age, birth weight for gestational age, low Apgar score (≤7) at 5/10 min, maternal educational level at childbirth, and maternal cohabitation status during pregnancy.